# Language outcomes of preschool children who are HIV-exposed uninfected: An analysis of a South African cohort

Freddy Green[1], Christopher du Plooy[2], Andrea M. Rehman[3], Raymond T. Nhapi[2,4], Marilyn T. Lake[2,5], Whitney Barnett[2,6], Nadia Hoffman[7], Heather J. Zar[2,5], Kirsten A. Donald[2,8], Dan J. Stein[7,8,9‡], Catherine J. Wedderburn[1,2,8‡]*

1 Department of Clinical Research, London School of Hygiene & Tropical Medicine, London, United Kingdom, 2 Department of Paediatrics and Child Health, Red Cross War Memorial Children's Hospital, University of Cape Town, Cape Town, South Africa, 3 MRC International Statistics & Epidemiology Group, Department of Infectious Disease Epidemiology, London School of Hygiene & Tropical Medicine, London, United Kingdom, 4 Division of Epidemiology and Biostatistics, School of Public Health and Family Medicine, University of Cape Town, Cape Town, South Africa, 5 South African Medical Research Council (SAMRC), Unit on Child & Adolescent Health, University of Cape Town, Cape Town, South Africa, 6 Department of Psychology and Human Development, Vanderbilt University, Nashville, Tennessee, United States of America, 7 Department of Psychiatry & Mental Health, University of Cape Town, Cape Town, South Africa, 8 Neuroscience Institute, University of Cape Town, Cape Town, South Africa, 9 SAMRC, Unit on Risk and Resilience in Mental Disorders, University of Cape Town, Cape Town, South Africa

‡ DJS and CJW are joint senior authors on this work.
* catherine.wedderburn@lshtm.ac.uk

## Abstract

### Introduction

There are approximately 16 million children who are HIV-exposed and uninfected (CHEU) worldwide. Studies suggest that CHEU are at risk for developmental impairment in infancy, particularly in language domains. However, there is limited research examining neurocognitive function in CHEU older than 2 years, including important pre-school years. This study aimed to investigate associations between HIV exposure without infection and neurocognitive outcomes and to determine risk factors for neurodevelopment in CHEU at age 3–4 years.

### Methods

The Drakenstein Child Health Study is a South African population-based birth cohort which enrolled women in pregnancy with ongoing follow up. Neurocognitive outcomes were assessed in children at 3.5 years by trained assessors blinded to HIV status including general cognitive function, language, and memory, measured using the Kaufmann Assessment Battery for Children, Second Edition (KABC-II). Data were compared between CHEU and children who were HIV-unexposed uninfected (CHUU) using multivariable logistic and linear regression, including testing for effect modification; sex-stratified risk factor analyses were performed.

**Data Availability Statement:** Data cannot be shared publicly because of ethical conditions with which study investigators are obliged to comply. Access to the project data is restricted to

nominated investigators approved by the University of Cape Town Human Research Ethics Committee, as per the consent document. Interested, qualified researchers may request to access this data by contacting the Drakenstein Child Health Study (via Lesley Workman, Senior Data Manager at lesley.workman@uct.ac.za) to submit a formal data use request and ensure required ethical approval received prior to use.

**Funding:** The Drakenstein Child Health Study was funded by the Bill & Melinda Gates Foundation [OPP 1017641], the National Institutes of Health (NIH) and the National Research Foundation, South Africa. Additional support for HJZ, DJS and KAD was provided by the South African Medical Research Council. An Academy of Medical Sciences Newton Advanced Fellowship (NAF002/1001) funded by the UK Government's Newton Fund supported the neurocognitive assessments. CJW was supported by the Wellcome Trust through a Research Training Fellowship [203525/Z/16/Z]. KAD is additionally supported by the NRF, the National Institute on Alcohol Abuse and Alcoholism (NIAAA) via (R21AA023887), the Collaborative Initiative on Fetal Alcohol Spectrum Disorders (CIFASD) developmental grant (U24 AA014811), and the US Brain and Behaviour Foundation Independent Investigator grant (24467). AMR is additionally supported by the UK Medical Research Council (MRC) and the UK Department for International Development (DFID) under the MRC/DFID Concordat agreement which is also part of the EDCTP2 programme supported by the European Union grant reference (MR/R010161/1). WB is supported by the SAMRC National Health Scholars programme. The funders had no role in study design, data collection and analysis, decision to publish, or preparation of the manuscript.

**Competing interests:** DJS has received research grants and/or consultancy honoraria from Discovery Vitality, Johnson & Johnson, Kanna, L'Oreal, Lundbeck, Orion, Sanofi, Servier, Takeda and Vistagen. All other authors report no competing interests. This does not alter our adherence to PLOS ONE policies on sharing data and materials.

## Results

A total of 497 children were included (97 [20%] CHEU; 400 [80%] CHUU; 50% male), with a mean age of 3.5 years (range 3.4–3.6). Groups had similar birth and household characteristics, although mothers of CHEU were older, on average. Overall, CHEU had lower expressive language scores compared to CHUU on unadjusted and adjusted analyses (effect size: -0.23 [95% CI -0.45, -0.01]). There were no group differences in general cognitive or memory function (p>0.05). On sex-stratified analyses, male CHEU were found to have higher odds of suboptimal cognitive development compared to male CHUU (aOR 2.28 [95% CI 1.06, 4.87], p = 0.034). Several other factors including birthweight, maternal education, maternal ART duration and HIV viral load during pregnancy were associated with cognition, memory, or expressive language outcomes in CHEU, dependent on child sex.

## Interpretation

The findings suggest that perinatal HIV exposure continues to be associated with impaired language development across the preschool years, highlighting the importance of targeting early interventions to optimise language outcomes. Further, the results suggest the importance of demographic, biological and HIV-related variables influencing developmental outcomes in CHEU. The greater risk of suboptimal cognitive development in male CHEU requires investigation around sex-specific mechanisms.

## Introduction

The number of people living with HIV remains high, estimated at 39 million globally [1]. Adolescent girls and young women remain disproportionately affected by HIV, accounting for 60% of new HIV infections in that age group [2]. In 2022 sub-Saharan Africa, adolescent girls and young women accounted for over 77% of new infections among 15–24 years olds [3]. However, over the past two decades there has been huge success in preventing vertical transmission of HIV with the scale up of antiretroviral therapy (ART) during pregnancy. Therefore, most children born to women living with HIV are HIV-exposed but remain uninfected; this population is estimated at 16 million worldwide and represents over 20% of children born in some countries, including South Africa [1].

Children who are HIV-exposed and uninfected (CHEU) are increasingly recognised as being vulnerable to poor health outcomes [4]. Through the Sustainable Development Goals there has been a shift to focus on ensuring that children not only survive, but also thrive. Previous reviews have found that CHEU have lower scores across many neurocognitive domains when compared with HIV-unexposed uninfected children (CHUU) [5, 6], and language has been noted to be a specific area of risk [7]. A recent meta-analysis of eight large studies of child neurodevelopment found that CHEU have particular risk of deficits in expressive language and gross motor function compared to CHUU by 24 months [8]. However, meta-analysis was not possible at older ages due to the scarcity of studies.

There is limited available research on cognition in CHEU older than 24 months [5]. One study in Malawi and Uganda using the Kaufman Assessment Battery for Children, 2nd edition (KABC-II) reported similar cognitive outcomes between CHEU and CHUU from infancy up to five years [9]. Comparable findings were seen in school-age children [10]. In Cameroon, authors found differences in KABC-II composite scores in school-aged CHEU and CHUU, although these differences were no longer apparent after adjusting for contextual factors [11].

However, only a few studies have included language outcomes in this age range. Researchers from the Democratic Republic of Congo reported differences between language impairment in CHEU compared to CHUU at 18–72 months [12]. Separately, a study of 2–12 year olds in Cambodia and Thailand also found poorer verbal ability, along with lower IQ, in CHEU [13]. However, many of these studies were conducted prior to the roll out of ART for all pregnant women, and differences in populations and study quality mean that further work is needed to understand neurocognitive outcomes in pre-school and school-age CHEU.

Several risk factors for poor neurocognitive development are highlighted in the literature and may impact upon the HIV exposure-outcome relationship [14]. Prior studies have demonstrated the importance of understanding these contextual factors [15] and having an adequately matched group of unexposed children for comparison [8]. The Drakenstein Child Health Study (DCHS) is a multidisciplinary South African population-based birth cohort, which was established to investigate early-life determinants of child health in a low socioeconomic peri-urban area of the Western Cape [16–18]. This provides a unique opportunity to investigate developmental outcomes on a larger sample than other recent studies in this context, providing access to comprehensive datasets. Previous research in this cohort found a high prevalence of developmental delay [14], and further, that CHEU had poorer receptive and expressive language outcomes at 24 months compared to CHUU [8]. Therefore, this analysis aimed to build upon these findings by assessing neurocognitive outcomes at 3.5 years comparing CHEU with CHUU from the DCHS and investigating the contextual predictors of neurodevelopment in CHEU.

## Methods

### Study design and setting

The study cohort is part of the DCHS. Pregnant women were recruited to the DCHS between 5 March 2012 and 31 March 2015 at 20–28 weeks' gestation at routine antenatal visits in two public healthcare clinics; mother-child dyads are being followed up in the DCHS with regular visits in addition to routine care. There is little emigration or immigration from or to the community, with strong retention in study follow up [19]. Antenatal HIV prevalence within this cohort was 21%, however, only two children acquired HIV infection from vertical transmission [20]. Inclusion criteria were 1) over 18 years of age, 2) receiving antenatal care at one of the participating clinics and 3) confirmed to be staying in the area for at least the next year. Written informed consent was obtained from all mothers with ongoing follow up of mothers and children. At 3.5 years neurocognitive assessments were conducted of all available children. This study was approved by the University of Cape Town Faculty of Health Sciences Human Research Ethics Committee.

### HIV exposure definition

Exposure, maternal HIV status, was confirmed through routine testing as per Western Cape prevention of mother-to-child transmission (PMTCT) guidelines on enrolment to the study. HIV-exposed children received HIV testing at 6 weeks, 9 months, 18 months and post-cessation of breastfeeding. CHEU were defined as those children born to mothers living with HIV but who remained HIV-negative, whereas CHUU children were born to HIV-negative mothers. Over the course of enrolment, provincial PMTCT guidelines changed, so triple ART initiation was dependent on initial CD4 count and clinical status for mothers enrolled before 2013, and mothers who did not fulfil the criteria received zidovudine prophylaxis; if enrolment was 2013 or later there was universal triple ART access. All HIV-exposed children received

prophylaxis at birth. CD4 and viral load counts were accessed from folder reviews and the National Health Laboratory Service. Children diagnosed with HIV were excluded from this analysis.

### Outcome definition (neurocognitive function)

**Measurement.** Neurocognitive function was measured using the KABC-II [21, 22]. The KABC is validated tool used globally to assess cognitive function in children from 3–18 years and has been used in children with HIV infection [23] and validated in South Africa [24, 25]. The tool was developed from neuropsychological theory and focuses on the processes needed to solve problems rather than their content. At 3.5 years, a full neurocognitive assessment was performed on all children available in the cohort by trained research assistants with the aid of interpreters where necessary. The KABC-II instructions were forward and back translated into local languages and consensus of the translations were done with community members to ensure that the translations were appropriate. The test administrators were all trained psychologists with MA level degrees. All data were quality controlled and checked, and missing data were investigated by test administrators.

**Domains.** The outcomes at 3.5 years are listed in S1 Table. General cognitive function, language, and memory domains were carefully selected to cover critical areas of child development. General cognitive function was measured using the four subtests listed in S1 Table. Raw scores for each test were converted to age-dependent scaled scores using the KABC-II handbook [22]. Scaled scores were then summed and converted to an age-dependent standard nonverbal index score (NVI) according to the KABC-II manual. Scores are standardised using normative data derived from a reference population to have a mean of 100 and a standard deviation of 15. An NVI of greater than one standard deviation or more below the mean is defined as suboptimal cognitive development [22].

Expressive language was measured using the Expressive Vocabulary subtest in the KABC-II. Memory was assessed using the Atlantis subtest within the KABC-II. Neither of the scores derived from the language or memory subtests can be dichotomised as the KABC-II recommend this only for global index scores such as NVI, so these outcomes are continuous, however, they were age-standardised.

### Demographic measures

Comprehensive data were collected using adapted questionnaires from the South African Stress and Health study at an antenatal home visit by trained study staff [26]. Demographic data were also collected at postnatal visits. Maternal psychosocial data were collected during the same antenatal visit. Depression was assessed using the Edinburgh Postnatal Depression Scale (EPDS) with a cut-off score of 13 or above. Alcohol use was assessed by the Alcohol, Smoking and Substance Involvement Screening Test, and smoking status using antenatal urine cotinine levels. Birth outcomes, such as birthweight and gestation were collected at delivery. Gestational age at delivery was assessed based on the second trimester ultrasound scan report; where these were not available, maternal report of last menstrual period dates and fundal height measurements were used. Low birthweight was defined as a birthweight of <2500g and prematurity as a gestation period of <37 weeks. Exclusive breastfeeding is recommended during the first six months of life, and we used this definition in this analysis.

### Statistical analysis

A complete-case analysis of mother-child pairs was performed, including children who completed all three outcome assessments and had non-missing values for *a priori* covariates.

Maternal and sociodemographic factors for the analytic cohort were expressed as frequencies and percentages for categorical data and means and standard deviations for continuous data. The summary statistics were shown for the total sample and across the two levels of exposure. Comparisons between CHEU and CHUU were made using two sample t-tests for continuous data and Fisher's exact tests and chi for categorical data respectively. Missing data were summarised for these factors, with their distribution across the exposure. A similar comparison was made between those children in the analytic cohort and the full cohort (n = 1141), to examine the distribution of covariates and assess for selection bias. A sensitivity analysis was also performed including all children who completed each assessment.

In order to assess the association between HIV exposure and neurocognitive development, potential confounders were identified by a directed acyclic graph (DAG) developed from the literature [7, 27–29] (S1 Fig). The minimal sufficient adjusted model indicated by the DAG for estimating the total effect of HIV exposure on neurocognitive development included child sex, maternal age at birth, maternal education and socioeconomic status. Linear regression was used to calculate mean difference estimates for the continuous outcomes, general cognitive function, expressive language and memory. General cognitive function was also dichotomised, and a logistic regression model was used to obtain estimates of association, odds ratios. Crude and adjusted odds ratios, their 95% confidence intervals and p-values were presented for group comparisons of general suboptimal cognitive development. Crude and adjusted mean differences, their 95% confidence intervals and p-values, and Cohen's d effect sizes were reported for cognitive function, language and memory; effect sizes were presented as a forest plot.

Effect modification was tested for each covariate remaining in the models using the likelihood ratio test. If a significant interaction was found using a likelihood ratio test, stratum-specific odds ratios/mean differences were presented in the results.

Finally, exploratory risk factor analyses were also performed on the variables identified by the DAG which may lie on the causal pathway. The associations of socioenvironmental variables, maternal psychological factors, birth outcomes, exclusive breastfeeding and HIV-related variables with the neurocognitive outcomes of CHEU were calculated, stratified by child sex given literature suggesting the role this plays in neurodevelopment [14, 29]. All analyses were performed using Stata version 15.1.

## Results

### Descriptive statistics

A total of 497 mother-child dyads were included in this analysis, 97 (19.5%) CHEU and 400 (80.5%) CHUU. Fig 1 shows a flow diagram of cohort enrolment and sample flow. Table 1 displays the characteristics within the analytic cohort, disaggregated by HIV exposure status. Mean group ages were 3.5 years across both groups (range 3.4–3.6). The groups were comparable in terms of socioeconomic factors, child sex and birth outcomes (birthweight and gestation). However, mothers with HIV tended to be older compared to mothers without HIV (mean 30.4 vs 26.4 years, p<0.001) and were less likely to have used alcohol during pregnancy (4.1% vs 14.3% p = 0.006). Data distributions were similar across the analytic cohort and the full cohort (S2 Table).

### General cognitive function

General cognitive scores were similar between CHEU (75.77) and CHUU (76.55), adjusted mean difference -0.72 (-4.06, 2.62) p = 0.672, effect size -0.04 (95% -0.27, 0.17). Similar

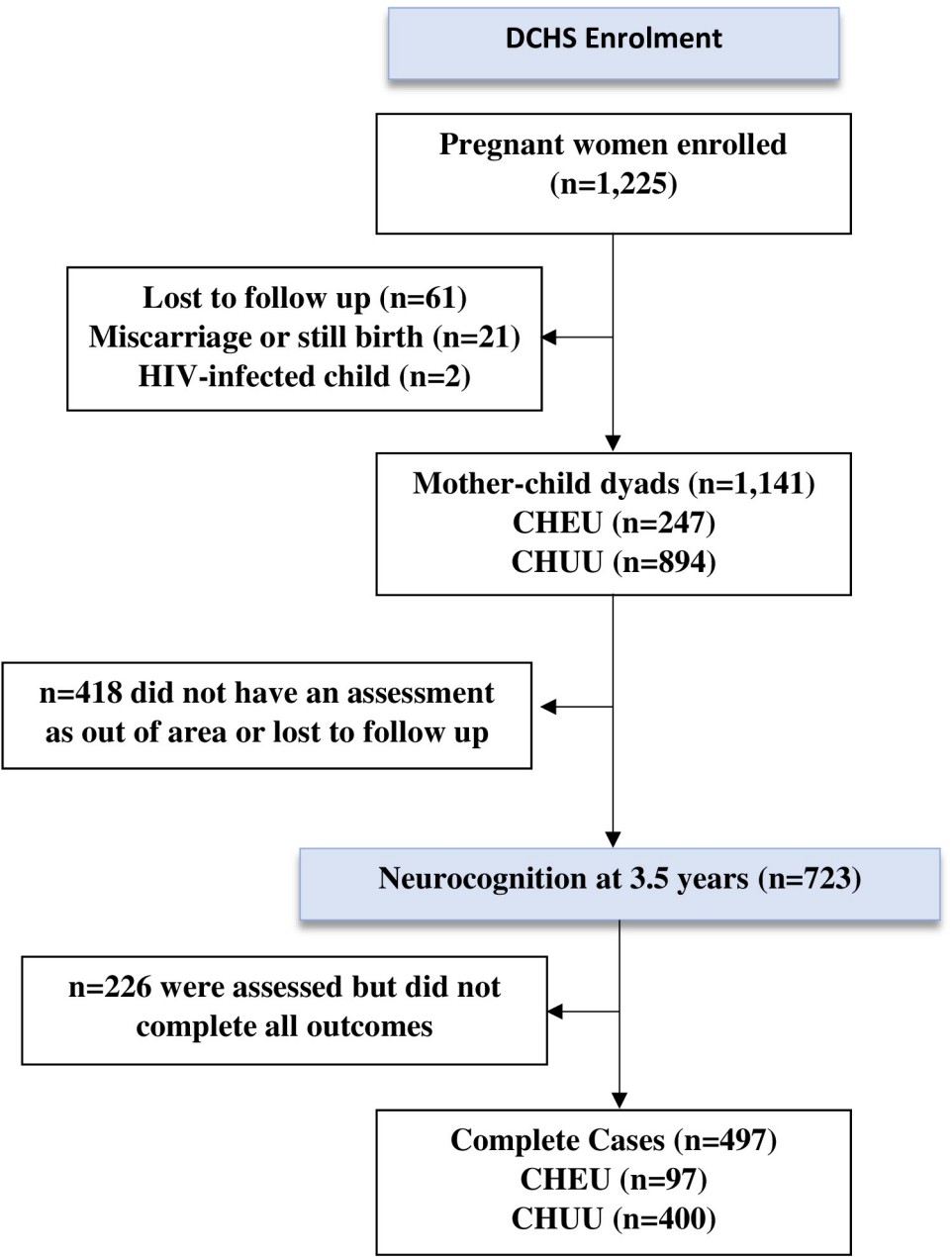

**Fig 1. DCHS cohort enrolment and sample used in this analysis.**

proportions of suboptimal cognitive development were also seen on both unadjusted and adjusted analyses (adjusted odds ratio 1.20 [95% CI 0.73, 1.99], p = 0.474) (Fig 2; S3 Table).

### Expressive language

On unadjusted analysis, there was an association between HIV exposure and lower expressive language scores (Fig 2; S3 Table). On average, CHEU scored 6.73 (SD = 1.79) and CHUU scored 7.23 (SD = 2.03), with a mean difference of -0.50 (95% CI -0.94, -0.06, p = 0.026), effect size -0.25 (95% CI -0.47, -0.03). The mean difference held on adjusting for covariates (sex,

**Table 1. Maternal, child and household characteristics, by HIV exposure.**

| Variable | Total cohort (n = 497) | CHEU, N (%) (n = 97) | CHUU, N (%) (n = 400) | P-value |
|---|---|---|---|---|
| **Child Characteristics** | | | | |
| Male | 249 (50.1) | 52 (53.6) | 197 (49.3) | 0.441 |
| Female | 248 (49.9) | 45 (46.4) | 203 (50.8) | |
| Birthweight (g), Mean (SD) | 3053.7 (579.3) | 3040.5 (618.5) | 3056.9 (570.3) | 0.803 |
| Premature Birth | 65 (13.1) | 12 (12.4) | 53 (13.3) | 0.760 |
| **Maternal Characteristics** | | | | |
| Maternal age at birth (yrs), Mean (SD) | 27.16 (5.69) | 30.43 (5.30) | 26.36 (5.49) | <0.001* |
| Education | | | | |
| Primary | 34 (6.8) | 7 (7.2) | 27 (6.8) | 0.072 |
| Some Secondary | 268 (53.9) | 63 (65.0) | 205 (51.2) | |
| Completed Secondary | 166 (33.4) | 24 (24.7) | 142 (35.5) | |
| Any Tertiary | 29 (5.8) | 3 (3.1) | 26 (6.5) | |
| Employed | 128 (25.8) | 27 (27.8) | 101 (25.2) | 0.941 |
| Exclusive Breastfeeding for 6 months | 81 (16.3) | 9 (9.3) | 72 (18.0) | 0.098 |
| Alcohol use in pregnancy | 61 (12.3) | 4 (4.1) | 57 (14.3) | 0.006* |
| Maternal depression (EPDS threshold) | 99 (20.0) | 19 (19.6) | 80 (20.0) | 0.139 |
| Smoker status | | | | |
| Non-smoker | 111 (22.3) | 26 (26.8) | 85 (21.3) | 0.202 |
| Passive smoker | 212 (42.7) | 45 (46.4) | 167 (41.8) | |
| Active smoker | 162 (32.6) | 23 (23.7) | 139 (34.7) | |
| Maternal death in first 3.5 years | 5 (1.0) | 2 (2.1) | 3 (0.8) | 0.252 |
| Maternal ART regimen (CHEU only) | | | | |
| PMTCT AZT monotherapy | - | 9 (9.3) | - | N/A |
| First-line triple therapy | - | 79 (81.4) | - | |
| Second-line therapy | - | 8 (1.03) | - | |
| Maternal ART initiation | | | | |
| Pre-pregnancy | - | 39 (40.2) | - | N/A |
| During pregnancy | - | 58 (59.8) | - | |
| Maternal CD4 count in pregnancy, Mean (SD) | - | 501.9 (238.1) | - | N/A |
| Maternal viral load in pregnancy | | | | |
| ≤ 1000 copies/ml | - | 61 (62.9) | | N/A |
| >1000 copies /ml | - | 7 (7.2) | | |
| **Household Characteristics** | | | | |
| Household income | | | | |
| <R1000/m | 206 (41.2) | 41 (42.3) | 165 (41.3) | 0.921 |
| R1000-R5000/m | 234 (47.1) | 46 (47.4) | 188 (47.0) | |
| >R5000/m | 57 (11.8) | 10 (10.3) | 47 (11.7) | |
| Water | 346 (69.6) | 66 (68.0) | 280 (70.0) | 0.814 |
| Toilet | 315 (63.4) | 56 (57.7) | 259 (64.8) | 0.198 |

N (%) for categorical variables, mean (SD) for continuous variables.

*p<0.05.

P-values generated using chi-squared test for categorical variables and t-tests or Fisher's exact for continuous variables. Percentages calculated out of all data. Missing data: prematurity n = 2, breastfeeding n = 1, alcohol use n = 50, EPDS n = 50, smoking n = 12, water n = 1. In CHEU only variables, ART regimen n = 1, viral load n = 29, CD4 count n = 12, Abbreviations: CHEU: children who are HIV-exposed uninfected, CHUU: children who are HIV-unexposed uninfected; AZT: zidovudine

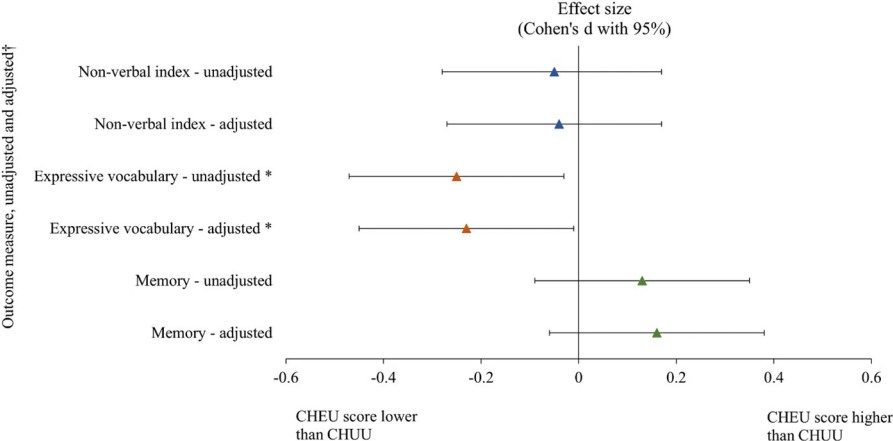

**Fig 2. Neurocognitive outcomes at 3.5 years compared between CHEU and CHUU.** *p<0.05. † Adjusted for child sex, maternal education, maternal age and household income. *p<0.05. CHEU: Children HIV-exposed uninfected, CHUU: Children HIV-unexposed uninfected.

maternal age at birth, maternal education and socioeconomic status) (-0.46 [95% CI -0.91, -0.02] p = 0.049) with an effect size of -0.23 (-0.45 to -0.01).

## Memory

There were no identified associations between HIV exposure and memory scores on unadjusted (p = 0.257) or adjusted analyses (p = 0.184), with an adjusted mean difference of 0.29 (95% CI -0.14, 0.71) and effect size 0.16 (95% CI -0.06, 0.38) (Fig 2; S3 Table).

A sensitivity analysis of each individual assessment including all children in the cohort who completed each test showed the CHEU/CHUU relationships held (S4 Table).

## Potential risk and protective factors

Likelihood ratio tests investigating effect modification on HIV exposure indicated an interaction between HIV exposure and the sex of the child (p = 0.008) for general suboptimal cognitive development. Examining each stratum for sex (Table 2), male CHEU had two times the (adjusted) odds of general suboptimal cognitive development compared to male CHUU (odds ratio of suboptimal cognitive development 2.28 (95% CI 1.06, 4.87, p = 0.034); there were no differences seen between female CHEU and CHUU (p = 0.239). There was a trend for a similar pattern examining cognitive continuous scores, but this was not significant (-3.35 [95% CI -8.06, 1.35]), and no evidence for effect modification in other development domains was detected using likelihood ratio tests.

An analysis of factors that may affect CHEU neurodevelopment was informed by the DAG. Fig 3 shows the results of the risk factor analyses stratified by sex. Overall, we found that higher maternal education (completion of secondary level) was associated with better memory in male CHEU, and better expressive language in female CHEU. Secondly, we found that higher birthweight was associated with improved cognitive function in male CHEU. Other factors, including household income, exclusive breastfeeding, maternal depression in pregnancy and prematurity, did not show evidence of an association with general cognitive function, language, or memory in CHEU in this analytic cohort. In terms of HIV-related variables, we found that initiation of ART before pregnancy (versus during pregnancy) was associated with

**Table 2. Sex-stratified unadjusted and adjusted analyses of neurocognitive outcomes at 3.5 years.**

| General Cognitive Function (NVI suboptimal development) | | | | |
|---|---|---|---|---|
| Sex/Exposure | Unadjusted odds ratio (95% CI) | p-value | Adjusted† odds ratio (95% CI) | p-value |
| **Female** | | | | |
| CHUU | 1 | 0.329 | 1 | 0.239 |
| CHEU | 0.72 (0.37, 1.39) | | 0.65 (0.32, 1.33) | |
| **Male** | | | | |
| CHUU | 1 | 0.072 | 1 | 0.034* |
| CHEU | 1.88 (0.94, 3.76) | | 2.28 (1.06, 4.87) | |
| **General Cognitive Function (NVI Score)** | | | | |
| Sex/Exposure | Unadjusted mean difference (95% CI) | p-value | Adjusted† mean difference (95% CI) | p-value |
| **Female** | | | | |
| CHUU | Reference | 0.691 | Reference | 0.454 |
| CHEU | 0.99 (-3.89, 5.86) | | 1.82 (-2.97, 6.61) | |
| **Male** | | | | |
| CHUU | Reference | 0.286 | Reference | 0.162 |
| CHEU | -2.37 (-6.73, 1.99) | | -3.35 (-8.06, 1.35) | |
| **Expressive language** | | | | |
| **Female** | | | | |
| CHUU | Reference | 0.147 | Reference | 0.227 |
| CHEU | -0.44 (-1.04, 0.16) | | -0.37 (-0.98, 0.23) | |
| **Male** | | | | |
| CHUU | Reference | 0.112 | Reference | 0.130 |
| CHEU | -0.52 (-1.17, 0.12) | | -0.54 (-1.23, 0.16) | |
| **Memory** | | | | |
| **Female** | | | | |
| CHUU | Reference | 0.766 | Reference | 0.718 |
| CHEU | 0.09 (-0.52, 0.71) | | 0.12 (-0.52, 0.75) | |
| **Male** | | | | |
| CHUU | Reference | 0.195 | Reference | 0.120 |
| CHEU | 0.35 (-0.18, 0.89) | | 0.46 (-0.12, 1.03) | |

Stratum-specific odds presented due to effect modification between sex and HIV (p = 0.008)

† Adjusted for child sex, maternal education, maternal age and household income.

*p<0.05.

CHEU: Children HIV-exposed uninfected, CHUU: Children HIV-unexpoed uninfected, NVI–Nonverbal Index

better expressive language in male CHEU, indicating greater ART duration was protective. Further, higher viral load (>1000 copies/ml versus <1000 copies/ml) in mothers during pregnancy was associated with worse cognitive function in female CHEU (Fig 3).

## Discussion

The findings of this cohort study indicate that children who are exposed to HIV but remain uninfected have poorer expressive language outcomes at 3–4 years, alongside similar cognitive and memory development, when compared with HIV-unexposed children. The analysis of general cognitive function suggested an interaction between sex and HIV exposure, and that male CHEU are at highest risk for suboptimal cognitive development. Finally, a sex-stratified analysis of risk factors in CHEU found that demographic (maternal education), biological

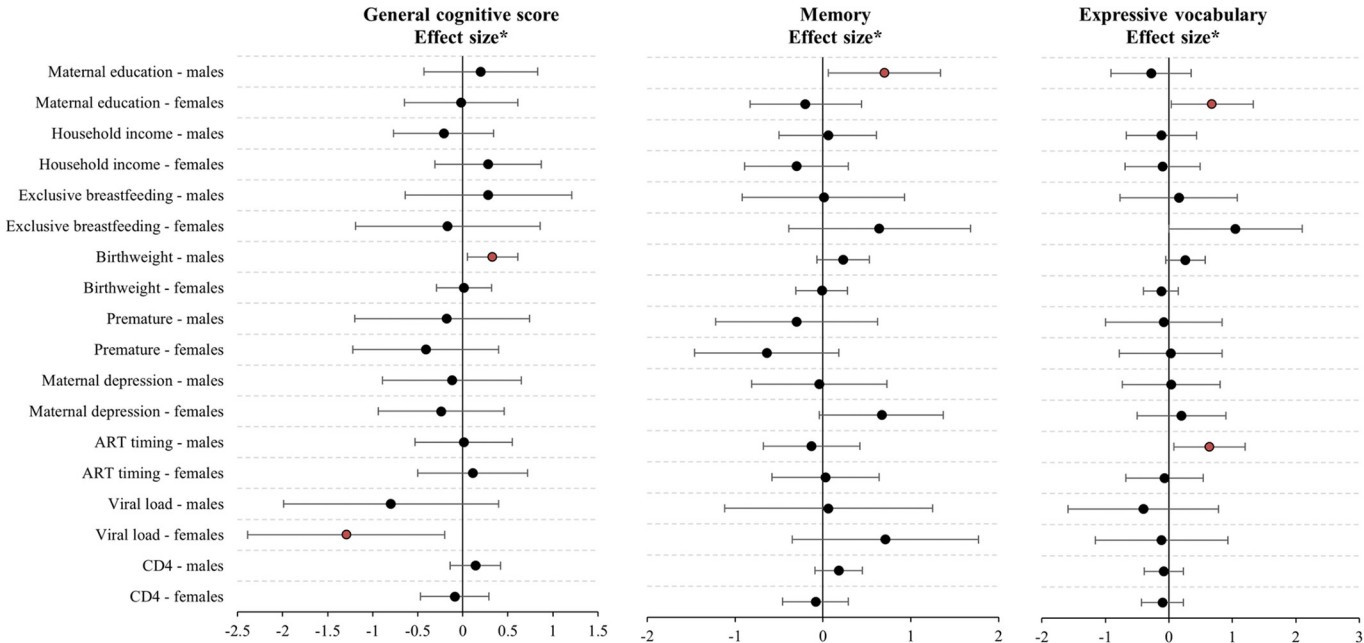

**Fig 3. Associations between risk and protective factors with neurocognitive development of CHEU.** Red colour indicates p<0.05. * Effect sizes for binary outcomes generated using Cohen's d. For continuous variables (birthweight and CD4 count), regression β values were generated. Factor information: Maternal education compares higher attainment (completed secondary/tertiary) versus primary/some secondary education; Household income compared higher income (>1000 rand/month) versus lower (<1000 rand /month); ART timing compared longer maternal ART (initiation pre-pregnancy versus during pregnancy); Viral load compared unsuppressed viral load (>1000 copies/ml) versus suppressed (<1000 copies/ml).

(birthweight) and HIV-related factors (maternal ART duration and viral load during pregnancy) were associated with CHEU neurocognitive outcomes, dependent on child sex.

The association between HIV exposure and poor expressive language development builds upon a previous analysis of the cohort at 24 months which found significant differences in language, with a very similar effect size (Cohen's d -0.23 [95%CI -0.41, -0.05]) [7]. Further, it supports a recent meta-analysis including eight studies which detected poorer scores in CHEU on expressive language compared to CHUU by 24 months [8]. The early years of life are a critical period for language development, and these findings suggest that early impairment, although subtle, persists during this time. Neurodevelopment affected by HIV exposure may show differences in language, either as this domain is most affected, or else differences are evident earlier than other domains. The complexity of language development may also mean that this domain is most affected in younger years. Language is known to be important for later school and academic outcomes [30], and there has been a report of ongoing language impairment in youth with HIV exposure [31]. These results suggest the importance of highlighting CHEU as a vulnerable group, detecting any delay in preschool language ability early, and informing the development of interventions to optimise later outcomes.

There was no association between HIV exposure and general cognitive function which is comparable to much of the recent literature, including a large study by Boivin *et al* implementing the KABC-II assessment [9]. However, these findings differ from the outcomes of previous systematic reviews [5, 6], potentially because studies included in the reviews generally examined younger children (under the age of 3 years), used tools that combined language and cognition, or were from the pre-ART era. Similarly, there was no evidence of an association between HIV exposure and memory at 3.5 years in this cohort. There is scarce research on

early memory development in CHEU, and this domain is often assessed as part of a composite score, and so further studies at older ages are required, particularly given higher order cognitive functions are still developing. Recent studies indicate memory and academic performance may be affected at older ages [15, 32], and so follow up is important.

Interestingly, there was evidence to suggest that male CHEU performed significantly worse than male CHUU. Prior research in this cohort has shown that boys perform worse than girls in early developmental assessments [14]. Similarly, the effects of sex have also been noted in other recent studies, suggesting male CHEU may be at most risk [15, 33]. Research set in the east Asia-Pacific region investigating sex disparities in child development in low- and middle-income country settings suggest that girls aged 3–5 years generally outperform boys on tests of development [34]. It is likely that environmental and psychosocial factors play a role in these differences, along with genetic and hormonal differences. Studies also suggests sexual dimorphism in early brain trajectories [35], and that there is significantly greater variance for several key brain structures in males, including white matter and cerebellar cortex, providing a novel perspective on sex differences in brain maturation [36]. Given reported differences in brain structure between CHEU and CHUU [37], further research using neuroimaging may help to better understand the underlying neuroanatomical pathways.

Investigating the determinants of CHEU neurodevelopment requires a holistic approach [29], including biological influences [38, 39], and environmental factors [40]. Several risk factors for poor neurocognitive development are highlighted in the literature and may impact upon the HIV exposure-outcome relationship [14]. The WHO Nurturing Framework [29] identifies five domains that are crucial to child development: health, nutrition, security, responsive caregiving and opportunities for early learning. The framework acknowledges that these domains are often dependent on socioeconomic status and immediate environmental factors, such as parental characteristics, as well as intrinsic factors such as child age and sex and that the accumulation of interacting factors may lead to the observed cognitive outcomes. Maternal education is frequently highlighted as important for child development [27], and we found an association between higher maternal education levels and better memory function in male CHEU, and expressive language in female CHEU. There is also considerable evidence that poor birth outcomes can adversely affect neurocognitive development [41]. In our exploration of other potential factors, we found an association between birthweight and cognitive outcomes in male CHEU, such that higher birthweight was associated with better outcomes. Birthweight has been found to be a protective factor [14], and conversely low birthweight is a known risk factor for morbidity [4], and may represent cumulative *in utero* insults including nutrition and infectious diseases [28, 42]. Our findings suggest low birthweight CHEU infants may be particularly vulnerable; it is likely that the combination of risk factors (low birthweight, male sex, and HIV exposure) may compound the risk of poor cognitive outcomes. Together, this research highlights the need to identify and protect those children with multiple risk factors. In terms of HIV-related variables, we found that better child outcomes were associated with longer duration of maternal ART (initiation pre-pregnancy versus during pregnancy). Conversely, worse outcomes were associated with high maternal viral load in pregnancy. This supports the existing literature where another South African study found maternal HIV viraemia in pregnancy predicted lower expressive language and motor outcomes and increased delay at 1 year [43]. Separately another study found infants born to mothers initiating ART during pregnancy had smaller subcortical volumes, compared to those initiating ART before conception and increasing ART exposure was protective [44]. Overall, these findings emphasise the importance of optimizing maternal HIV care in pregnancy to improve child outcomes.

This study adds value to the field with demographically comparable unexposed control group, providing a reasonable estimate of the counterfactual. This overcomes common methodological limitations that have hampered many prior studies, namely small sample sizes, unmeasured confounders and unbalanced exposure groups in the context-specific literature. The comprehensive data collected by the DCHS on maternal, environmental, and household characteristics allowed this analysis to control for factors with an established impact on neurocognitive outcomes in children. With a large enough sample size, these factors were included in the analysis without introducing data sparsity. This cohort may also be generalisable to other sub-Saharan contexts and uses the KABC, a well-validated tool [24]. These approaches lend greater specificity for measuring the impact of interventions focused on improving deficits in cognition [45], suggesting the results may be used to evaluate future interventions.

This study has some limitations. Although results held on adjusted analyses, other unmeasured potential confounders may have resulted in residual confounding. Further work needs to be carried out to understand the impact of other variables, including household composition, in the association between HIV exposure and neurocognitive development and to replicate the findings in other settings. While infant feeding practices are an important potentially modifiable factor for the neurocognitive growth in children [46], there was a low prevalence of exclusive breastfeeding in this cohort which may have limited the power to investigate an association with neurocognitive outcomes. A complete-case analysis was used to mitigate missing outcome data and potential selection bias was evaluated. It was noted that missing data was similar across exposure groups and the distribution of demographic and other covariate data for the full cohort was similar to the analytic cohort; results held on sensitivity analyses of each outcome measure. The KABC-II is normed using data from a standardised sample from a high-income setting which creates limitations in this setting, and it will be important to compare with other scales with contextually appropriate norms in the future. However, the tool has been validated in the sub-Saharan context [47], and in young African children affected by HIV [24] and we focused on between-group comparisons. There are also other components of neurocognition such as executive function, motor control and social cognition that were not reported here. Finally, we note that the cohort was recruited in 2012–2015. In considering comparisons to current times, new note that our cohort is representative of women living with HIV nationally [48], with similar sociodemographics [49], and overall ART coverage at the time of study enrolment was comparable to many SSA countries in recent times [1]. Further work is needed to assess generalisability, particularly given the relationship between efavirenz and neurocognition [50, 51] and the introduction of dolutegravir-based ART [52].

## Conclusions

This analysis builds upon previous research that identified an association between HIV exposure in children and language development in sub-Saharan Africa, showing continued subtle impairment in expressive language at 3–4 years. This is notable giving the rapidly growing and ageing population of CHEU [1], and indicates the need for strategies to detect language problems early and implement interventions to optimise outcomes. The identified vulnerability of CHEU and other biological and demographic risk factors may help target intervention strategies. Further research should be undertaken to assess whether these results can be replicated in other contexts, focusing on understanding the mechanisms behind adverse neurodevelopment outcomes and paying particular attention to potential sex differences.

## Supporting information

**S1 Fig. Directed acyclic graph displaying plausible pathways for the association between HIV exposure and neurocognitive development.**
(PDF)

**S1 Table. Outcome measures.**
(PDF)

**S2 Table. Maternal, child and household characteristics of the full cohort, by exposure.**
(PDF)

**S3 Table. Neurocognitive outcomes at 3.5 years compared between CHEU and CHUU in the analytic (complete-case) cohort.**
(PDF)

**S4 Table. Neurocognitive outcomes at 3.5 years compared between CHEU and CHUU among all those completing each assessment.**
(PDF)

## Acknowledgments

We thank the families and children who participated in this study. We gratefully recognise the study staff at Mbekweni and T.C. Newman clinics, the study data team and laboratory teams, and the clinical and administrative staff at Paarl Hospital for their support of the study. We acknowledge the advice from members of the study International Advisory Board and thank our collaborators.

## Author Contributions

**Conceptualization:** Heather J. Zar, Kirsten A. Donald, Dan J. Stein, Catherine J. Wedderburn.

**Data curation:** Freddy Green, Marilyn T. Lake, Catherine J. Wedderburn.

**Formal analysis:** Freddy Green, Andrea M. Rehman, Raymond T. Nhapi.

**Funding acquisition:** Heather J. Zar, Kirsten A. Donald, Dan J. Stein.

**Investigation:** Christopher du Plooy, Whitney Barnett, Nadia Hoffman, Catherine J. Wedderburn.

**Methodology:** Freddy Green, Christopher du Plooy, Heather J. Zar, Kirsten A. Donald, Dan J. Stein, Catherine J. Wedderburn.

**Project administration:** Whitney Barnett, Nadia Hoffman.

**Resources:** Marilyn T. Lake, Heather J. Zar, Kirsten A. Donald, Dan J. Stein.

**Supervision:** Catherine J. Wedderburn.

**Validation:** Christopher du Plooy, Andrea M. Rehman, Raymond T. Nhapi.

**Writing – original draft:** Freddy Green, Catherine J. Wedderburn.

**Writing – review & editing:** Freddy Green, Christopher du Plooy, Andrea M. Rehman, Raymond T. Nhapi, Marilyn T. Lake, Whitney Barnett, Nadia Hoffman, Heather J. Zar, Kirsten A. Donald, Dan J. Stein, Catherine J. Wedderburn.

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
