## [Decision Letter · Decision Letter 0]

11 Oct 2023

PONE-D-23-27426Neurocognitive and language outcomes of preschool children who are HIV-exposed uninfected: an analysis of a South African cohortPLOS ONE

Dear Dr. Wedderburn,

Thank you for submitting your manuscript to PLOS ONE. After careful consideration, we feel that it has merit but does not fully meet PLOS ONE’s publication criteria as it currently stands. Therefore, we invite you to submit a revised version of the manuscript that addresses the points raised during the review process.

We look forward to receiving your revised manuscript.

Kind regards,

Andrea L. Conroy, PhD

Academic Editor

PLOS ONE

Journal Requirements:

2. Please expand the acronym “NIAAA” (as indicated in your financial disclosure) so that it states the name of your funders in full.

3. Thank you for stating the following in the Competing Interests: 

"DJS has received research grants and/or consultancy honoraria from Discovery Vitality, Johnson & Johnson, Kanna, L’Oreal, Lundbeck, Orion, Sanofi, Servier, Takeda and Vistagen. All other authors report no competing interests."

We note that you received funding from a commercial source: Discovery Vitality, Johnson & Johnson, Kanna, L’Oreal, Lundbeck, Orion, Sanofi, Servier, Takeda and Vistagen.

Within this Competing Interests Statement, please confirm that this does not alter your adherence to all PLOS ONE policies on sharing data and materials by including the following statement: ""This does not alter our adherence to PLOS ONE policies on sharing data and materials.” (as detailed online in our guide for authors http://journals.plos.org/plosone/s/competing-interests).  If there are restrictions on sharing of data and/or materials, please state these. Please note that we cannot proceed with consideration of your article until this information has been declared. 

4. Please ensure that you refer to Figure 1 in your text as, if accepted, production will need this reference to link the reader to the figure.

**Additional Editor Comments:**

The authors are enthusiastic about the work and believe the study was well conducted, is important to the field, and is well written. They provide several constructive suggestions to improve the manuscript and offer suggestions on alternative analytic approaches to assess outcomes using the population of HIV unexposed uninfected children as population controls. I strongly suggest the authors incorporate the additional analytic approaches in the revised manuscript as a sensitivity analysis or supplement. At minimum, the limitations of the tools must be clearly described following the guidance outlined by reviewers.

Reviewers' comments:

Reviewer's Responses to Questions

**Comments to the Author**

1. Is the manuscript technically sound, and do the data support the conclusions?

Reviewer #1: Partly

Reviewer #2: Yes

Reviewer #3: Partly

2. Has the statistical analysis been performed appropriately and rigorously? 

Reviewer #1: No

Reviewer #2: Yes

Reviewer #3: Yes

3. Have the authors made all data underlying the findings in their manuscript fully available?

Reviewer #1: No

Reviewer #2: Yes

Reviewer #3: Yes

4. Is the manuscript presented in an intelligible fashion and written in standard English?

Reviewer #1: Yes

Reviewer #2: Yes

Reviewer #3: Yes

5. Review Comments to the Author

Reviewer #1: This paper assesses general cognitive ability, language, and memory in South African children aged 3.5 years who were exposed to HIV in utero, but who were uninfected themselves, as compared to children who were HIV-unexposed and uninfected. All children were part of a birth cohort enrolled in the Drakenstein Child Health Study, and all data were collected when children were 3.5 years of age between March 2012 and March 2015. Assessments were made using the Kaufman Assessment Battery for Children, 2nd edition (KABC-II). The paper is well written and clear, and the statistical analyses for the most part support the conclusions. However, there are important gaps in context and methods that need to be integrated into the paper before full assessment of the validity and generalizability of the paper’s findings can be made.

Questions/concerns

1) The data were collected between 2012-2015 and are between 8 and 11 years old. The results are presented as those that “continue to build” on findings between HIV exposure and language development, but the generalizability of the findings in the vastly changed landscape of HIV treatment and sociodemographic shifts since then are not addressed. Basic information from the time of collection vs. current day about mother-to-child transmission stats, types of ART used (including use of efavirenz which can affect neurocognitive outcomes), and home environment is not included or discussed as a limitation. Has HIV family structure shifted over the past decade? Are more children exposed to HIV now living in two-parent households and thus potentially have higher income, better food security, and greater access to healthcare? How does this differ from 2012-2015? These important differences in context are not well described but are critical for full interpretation and applicability of the findings.

2) It’s unclear if the KABC-II has been validated in South Africa. The authors cite a meta-analysis of its use in multi-cultural settings, but its use specifically in South Africa where there is a wide range of demographics are not addressed. If it has not been validated in South Africa, this should be included as a limitation in the discussion.

3) There are important gaps in the description of testing. More than 30 languages are spoken in South Africa. Was the test available in all languages spoken in the region in which the women were enrolled? What are the limitations if full translation was not possible? What was the education/training level of the test administrators?

4) What was the rationale for testing for language at 3.5 years? This wasn't clearly stated in the manuscript. Typically, language development is divided into 3-12 months, 12-18months, 18months-2 years, 2-3 years, 3-5 years.

5) There is not statistical testing for multiple comparisons in the risk factor analysis. Multiple risk factors compared against multiple scales on the test require this testing.

Reviewer #2: This is an incredibly important area of research and I commend the authors for their work. This manuscript was clear, concise, and very enjoyable to read. I believe this work makes a meaningful, and very much needed, contribution to the academic research in this field.

Introduction

Might be worth highlighting in the introduction that the majority (somewhere around 80%) of women living with HIV are of reproductive age. Intro skips from people living with HIV to children being born to HIV-infected mothers, and it might add to the argument to focus in on the large number of people living with HIV who are women of reproductive age (to further emphasize the importance of the issue the authors are examining here).

Paragraph 2: suggested editing to “previous reviews of work in LMICs”, to clarify that the comparison between CHEU and unexposed children is all with LMICs (if this is true), or remove the LMICs framing if these meta-analyses included HICs

Specify definition of “young” for “young CHEU” (paragraph 2), or refer to all as children <24 months

Specify “older ages” in paragraph 3, e.g. children over 24 months? (since young or older could be relative), even “pre-school and school-aged children” to be a bit more specific about age

Paragraph 3, suggested adding parameters to “understand CHEU outcomes” e.g. “to understand neurocognitive outcomes in pre-school and school-aged CHEU children”

Paragraph 4, suggest adding parameters to “Several risk factors for development” e.g. “risk factors for poor neurocognitive development”

Paragraph 4, suggested editing “such as maternal characteristics” to “parental characteristics” (to emphasize the importance of both parents participating in caregiving for children)

Suggest that paragraph 4 of the introduction could be included in the discussion, around parameters outside of HIV status that intersect with the neurocognitive outcomes here, to shorten the intro a bit

Suggested editing “Drakenstein Child Health Study” to include “the study cohort is part of the Drakenstein Child Health Study”, to clarify for the reader that’s why you’re discussing this cohort, or clarify is this cohort was established specifically for this study?

Suggest reframing “harnessing the richness of the data collected to assess potential confounding variables” to “providing access to comprehensive datasets” (as “richness” could be seen as subjective and statistical methodology should determine confounding variables, vs. assessing potential)

Methods

This appears to be a sub-analysis of the broader DCHS study (e.g. references 20,21), would clarify in the methods how participants were enrolled in the broader study, even overall aims of the broader study, to put this analysis in context

Clarify statement “mother-child dyads are being followed up.” e.g. prenatal visits across pregnancy followed regional recommendations etc.

Suggested removing “so the population is relatively stable”

Keep past tense consistent (e.g. HIV prevalence in this cohort was 21%)

Does “only two children have HIV infection” mean from vertical transmission?

Suggested editing Exposure Definition to “HIV Exposure Definition”

Could provide a bit more detail around ARV treatment (e.g. were all women receiving the same medications? Is there data on what women received which medications if there are multiple ARV regimens prescribed, what prophylactic therapy was given to children at birth? Since there is some evidence that drug exposures have impact it may be important information for the reader to have). In addition, if mothers are receiving various treatments this should likely be included as a covariate (likely too small a cohort to examine this much further).

Since the children here don’t have HIV infection it’s likely more important to clarify that the neurocognitive assessment tools used here have been validated in these populations (e.g. culturally appropriate, similar geography, validation with younger cohorts (since age range is 3-18 and this cohort is all within the younger range etc.)

Under “Domains”, would clarify if these are validated approaches to data analysis (could also be stated in the statistical methods section), or if all methods described (beyond establishing age-dependent scores) are based on what’s outlined in the KABC handbook

Would specify how gestational age at delivery was assessed (e.g. by ultrasound?)

Under statistical analysis, could expand on what a “complete case analysis is” (e.g. is this a separate analysis of each invidual dyad?)

Statistical analysis states “children who completed all three outcome assessments”, does that mean any child who didn’t complete all assessments were removed from the analysis? Would include that if that’s the case, and/or clarify how the analysis accounted for missing data

Results

Suggest removing “split between” as it might imply assignment to groups

Suggest removing “complete-case cohort” as could imply this is subset of the complete cohort (e.g. a case-cohort), would just refer to this as the study cohort

Would it be possible to include a Figure to present the data? While the tables are very comprehensive a figure with the cognitive data might be a nice visual presentation for the reader

Discussion

Suggest including some discussion around why language is impacted and no other parameters? Is there a pathobiology hypothesis there (e.g. developmental processes in utero critical to language that may be differentially impacted?), or is this an age-specific observation? Or perhaps these methodology we have available for these types of assessments is most sensitive in terms of picking up differences in language?

Maybe expand on the finding that LBW was associated with poorer cognitive function in CHEU, perhaps compounding risk factors, highlighting the need to protect these children. E.g. expand on the sentence “Our findings suggest low birthweight CHEU infants may be particularly vulnerable.”

Suggest removing “large sample size”, as this could be perceived as subjective (likely could also change “small sample size” to insufficient sample size, same with “large enough sample size”, would suggest “sufficient power, or sample size”)

Suggest removing “unlike in many of the studies performed to date” and leave as “allowing this analysis to control for factors with an established impact on neurocognitive outcomes in children”

Suggested editing “this analysis has limitations” to “this study”, as limitations are not restricted to the analysis

“There were some missing outcome data across the cohort which may have resulted in selection bias.” Is stated in the discussion, should be expanded upon in the methods section. Suggest the section on missing data in the limitations paragraph of the discussion could be moved to the methods section (since it’s great to highlight this as a limitation but that’s true of essentially every study and just requires transparency in the methods on how it’s accounted for).

Conclusion

Suggested rephrasing “continued subtle impairment” as subtle could be somewhat objective, perhaps consider removing it as the analysis did how an impairment

Would caution the authors around this statement “The identified vulnerability of male CHEU, and low birthweight as a risk factor, may help target intervention strategies.” As it could imply sex-stratification of resource allocation, which, given female children often suffer from lower resource allocation, could send the wrong message … arguably the results in entirety show that all children who are CHEU require resource allocation (which would already be a targeted population for intervention)

Reviewer #3: It was a pleasure reviewing this manuscript that attempted to determine HIV associated neurocognitive impairment in CHEU at 3.5 years given that previous studies had covered a younger age group. There are a few comments below that need to be addressed to improve the manuscript.

Major

1. KABC raw scores were converted to scaled scores using norms from the manual and then converted to standard scores. The manual norms are from US samples. This implies that the South African sample were compared to US norms which may give biased results given cultural differences (where test items maybe more familiar to 3 to 4 year old US children than SA children). This inflates the actual rate of impaired scores and may explain why there was high rate of impairment in the CHUU of 63.5%. This high rate of impairment in a ‘control group’ should raise a red flag. What the authors can do is to use the CHUU to generate ages adjusted Z scores which can be used as the outcomes and not the scores derived from US norms. In this way, CHEU children will be compared to their our ‘norms’.

2. The high rate of impairment needs to be addressed in the discussion. What can explain these high figures in both groups?

3. Memory was assessed using Atlantis which as per the KABC manual is a learning measure. There are other memory tests like Hands Movement and Number Recall which would be more appropriate.

4. The above comments if not addressed need to be included under study limitations.

Minor

1. Why do the authors emphasise ‘language’ in the title when they also state ‘neurocognition’ This appears like an over emphasis given both outcomes were assessed using the same battery. Since language was the primary focus, they could remove neurocognition (which was not associated with HIV in the study anyway) and be direct with using language only.

2. It would be helpful for the reader if the age range of the children was given in the methods of the abstract as well as the period when the study was done (word limit permitting).

3. In the abstract, the KABC battery used is the ‘second edition’. This should be stated to distinguish it from the original version.

4. The comparison group should be stated in the methods of the abstract not in the results.

5. The Bayley Scales have been used more widely in Sub-Saharan Africa than the KABC-II in children 3 to 4 years old. It is not clear why the KABC was chosen instead. It does assess from 3 to 18 years, but some authors have found challenges using it in children less than 5 years due to item difficulty in younger African children. This is alluded to in the major comments above concerning the large impairment rate.

6. The KABC-II gives two summary scores, the NVI which the present study used and the Mental Processing Composite which is a measure of overall cognition. Why did the authors choose

the NVI over MPI, want advantages does the former possess?

7. This sentence in the statistics section needs to be corrected; ‘The summary statistics were shown for the full cohort and across the two levels of exposure with comparisons using two sample t-tests or Fisher’s exact tests and chi square tests for categorical and continuous data respectively.’

8. The numbers in figure 1 do not add up, this needs to be corrected.

6. PLOS authors have the option to publish the peer review history of their article (what does this mean?). If published, this will include your full peer review and any attached files.

Reviewer #1: No

Reviewer #2: No

Reviewer #3: No

---

## [Author Response · Author response to Decision Letter 0]

23 Dec 2023

21 December 2023

Dr Andrea L. Conroy

Academic Editor

PLOS ONE

Dr Conroy,

Thank you for considering our manuscript titled “Language outcomes of preschool children who are HIV-exposed uninfected: an analysis of a South African cohort” (reference: Submission ID PONE-D-23-27426) for publication in PLOS ONE and for the opportunity to revise and resubmit.

We thank the Editor and the reviewers for the detailed and thoughtful comments. We have updated the manuscript in response to these comments and feel that it is much improved as a result. We have carefully addressed each comment; please find our point-by-point response below, which also indicates the changes made to the manuscript.

Yours sincerely,

Dr Catherine Wedderburn

On behalf of all the authors

 

Editor Comments to Author:

Journal Requirements:

Response: Thank you, we have reviewed this.

2. Please expand the acronym “NIAAA” (as indicated in your financial disclosure) so that it states the name of your funders in full.

NIAAA: National Institute on Alcohol Abuse and Alcoholism

Response: We have included this in the paper and will add to the cover letter.

3. Thank you for stating the following in the Competing Interests: 

"DJS has received research grants and/or consultancy honoraria from Discovery Vitality, Johnson & Johnson, Kanna, L’Oreal, Lundbeck, Orion, Sanofi, Servier, Takeda and Vistagen. All other authors report no competing interests."

We note that you received funding from a commercial source: Discovery Vitality, Johnson & Johnson, Kanna, L’Oreal, Lundbeck, Orion, Sanofi, Servier, Takeda and Vistagen.

Within this Competing Interests Statement, please confirm that this does not alter your adherence to all PLOS ONE policies on sharing data and materials by including the following statement: ""This does not alter our adherence to PLOS ONE policies on sharing data and materials.” (as detailed online in our guide for authors http://journals.plos.org/plosone/s/competing-interests). If there are restrictions on sharing of data and/or materials, please state these. Please note that we cannot proceed with consideration of your article until this information has been declared. 

Response: Thank you for highlighting this. As per our email correspondence, we have edited the statement to read as follows: 

“DJS has received research grants and/or consultancy honoraria from Discovery Vitality, Johnson & Johnson, Kanna, L’Oreal, Lundbeck, Orion, Sanofi, Servier, Takeda and Vistagen, none of which related to the current research. All other authors report no competing interests. This does not alter our adherence to PLOS ONE policies on sharing data and materials. There are no patents, products in development, or marketed products associated with this research to declare.”

4. Please ensure that you refer to Figure 1 in your text as, if accepted, production will need this reference to link the reader to the figure.

Response: Apologies for this oversight, we have added in a reference.

Response: Thank you, we have done this.

Additional Editor Comments:

The authors are enthusiastic about the work and believe the study was well conducted, is important to the field, and is well written. They provide several constructive suggestions to improve the manuscript and offer suggestions on alternative analytic approaches to assess outcomes using the population of HIV unexposed uninfected children as population controls. I strongly suggest the authors incorporate the additional analytic approaches in the revised manuscript as a sensitivity analysis or supplement. At minimum, the limitations of the tools must be clearly described following the guidance outlined by reviewers.

Response: Many thanks for the helpful comments, we have responded to them below and have updated the manuscript and thank the editor and reviewers. We have looked into additional analytic approaches and also adding in tool limitations. We respond in full to the reviewers below, but would like to highlight that we have: (1) investigated the suggested approaches and what may be possible and include a detailed explanation in relevant sections; (2) we have updated our analyses to include an emphasis on effect sizes, rather than purely statistical significance; (3) we have reduced the focus on impairment, ensured we use the terminology ‘suboptimal development’ and instead have shifted to group comparisons; and (4) we have expanded our analyses and stratified the risk factor analysis by sex based on our earlier findings and the increased emphasis on this approach in the literature. We also carefully describe the tool limitations in the Discussion. 

Reviewers' comments:

Reviewer's Responses to Questions

Comments to the Author

1. Is the manuscript technically sound, and do the data support the conclusions?

Reviewer #1: Partly

Reviewer #2: Yes

Reviewer #3: Partly

Response: Many thanks, we hope we have responded to the comments adequately, and the paper is stronger as a result. 

2. Has the statistical analysis been performed appropriately and rigorously? 

Reviewer #1: No

Reviewer #2: Yes

Reviewer #3: Yes

Response: Many thanks, we have now improved the analyses per Reviewer #1 comments. 

3. Have the authors made all data underlying the findings in their manuscript fully available?

Reviewer #1: No

Reviewer #2: Yes

Reviewer #3: Yes

Response: Thank you. We are unable to make the data publicly available in a repository for ethical reasons, and have updated our statement as below. We are happy to provide more information as required, and will respond to any request in a timely manner.

“Data cannot be shared publicly because of ethical conditions with which study investigators are obliged to comply. Access to the project data is restricted to nominated investigators approved by the University of Cape Town Human Research Ethics Committee, as per the consent document. Interested, qualified researchers may request to access this data by contacting the Drakenstein Child Health Study (via lesley.workman@uct.ac.za) to submit a formal data use request and ensure required ethical approval received prior to use.”

4. Is the manuscript presented in an intelligible fashion and written in standard English?

Reviewer #1: Yes

Reviewer #2: Yes

Reviewer #3: Yes

Response: Thank you.

5. Review Comments to the Author

Reviewer #1: This paper assesses general cognitive ability, language, and memory in South African children aged 3.5 years who were exposed to HIV in utero, but who were uninfected themselves, as compared to children who were HIV-unexposed and uninfected. All children were part of a birth cohort enrolled in the Drakenstein Child Health Study, and all data were collected when children were 3.5 years of age between March 2012 and March 2015. Assessments were made using the Kaufman Assessment Battery for Children, 2nd edition (KABC-II). The paper is well written and clear, and the statistical analyses for the most part support the conclusions. However, there are important gaps in context and methods that need to be integrated into the paper before full assessment of the validity and generalizability of the paper’s findings can be made.

Response: Thank you.

Questions/concerns

1) The data were collected between 2012-2015 and are between 8 and 11 years old. The results are presented as those that “continue to build” on findings between HIV exposure and language development, but the generalizability of the findings in the vastly changed landscape of HIV treatment and sociodemographic shifts since then are not addressed. Basic information from the time of collection vs. current day about mother-to-child transmission stats, types of ART used (including use of efavirenz which can affect neurocognitive outcomes), and home environment is not included or discussed as a limitation. Has HIV family structure shifted over the past decade? Are more children exposed to HIV now living in two-parent households and thus potentially have higher income, better food security, and greater access to healthcare? How does this differ from 2012-2015? These important differences in context are not well described but are critical for full interpretation and applicability of the findings.

Response: Thank you for raising these important points. We have carefully considered the question of generalisability and have added more detail to the methods and discussion accordingly. 

Generally, we are not aware of major sociodemographic shifts in this region. We recognise the changing HIV/ART guidelines in recent years and our cohort of pregnant women were recruited in 2012-2015 with 3.5 year neurocognitive testing from 2016–2018. However, we feel it is comparable to current times. Based on the South African General Household Survey, 39.2% completed secondary education or higher which is comparable to the rest of South Africa where 46% of people had ≥grade 12 qualification in 2019 [1]. In terms of HIV, South Africa has the highest numbers of children who are HEU worldwide, and over 21% of children are born HEU each year [2]. Similarly, antenatal HIV prevalence in the DCHS was 21% across the cohort [3], and maternal CD4 cell counts in pregnancy parallel a national study of women with HIV [4]. Coverage of pregnant women who receive ARVs for PMTCT has been high since 2013, with minimal change subsequently [5], indicating these data continue to be relevant. These results provide support that our data are representative of the rest of South Africa in more recent times. We have added in references and expanded our discussion. 

However, we acknowledge that since our study the ART guidelines have changed and new WHO guidelines (WHO, 2021) recommend dolutegravir-based ART as first-line for women of child-bearing age. Therefore, we feel that our children from 2012–2015 are generally representative of children who are HEU up until the guideline change to recommend dolutegravir. The transition to dolutegravir has been slower than anticipated, and the COVID epidemic has negatively impacted HIV services and socioeconomic factors from across the world. We are aware of new studies being undertaken in this area which will hopefully be able to expand the literature on dolutegravir, including the Dolphin-2 trial, however, these will still be a few years to produce outcomes. Overall, the period during which the children in our study were exposed to HIV included policies and access to ART which are comparable to current times and are important to illustrate the outcomes of the millions of HEU children growing up in the pre-dolutegravir era. We have added cohort age as a consideration, and outline how it is still relevant on page 12. 

“Finally, we note that the cohort was recruited 2012-2015. In considering comparisons to current times, we note that our cohort is representative of women living with HIV nationally [4], with similar sociodemographics [1], and overall ART coverage at the time of study enrolment was comparable to many SSA countries in recent times [5]. Further work is needed to assess generalisability, particularly given the relationship between efavirenz and neurocognition [6, 7] and the introduction of dolutegravir -based ART [8].” 

2) It’s unclear if the KABC-II has been validated in South Africa. The authors cite a meta-analysis of its use in multi-cultural settings, but its use specifically in South Africa where there is a wide range of demographics are not addressed. If it has not been validated in South Africa, this should be included as a limitation in the discussion.

Response: Thank you, the KABC-II has been validated in South Africa. We have added two references below.

Chernoff, M. C., Laughton, B., Ratswana, M., Familiar, I., Fairlie, L., Vhembo, T., ... & Boivin, M. J. (2018). Validity of neuropsychological testing in young African children affected by HIV. Journal of Pediatric Infectious Diseases, 13(03), 185-201

Mitchell, J. M., Tomlinson, M., Bland, R. M., Houle, B., Stein, A., & Rochat, T. J. (2017). Confirmatory factor analysis of the Kaufman assessment battery in a sample of primary school-aged children in rural South Africa. South African Journal of Psychology, 48(4), 434-452.

3) There are important gaps in the description of testing. More than 30 languages are spoken in South Africa. Was the test available in all languages spoken in the region in which the women were enrolled? What are the limitations if full translation was not possible? What was the education/training level of the test administrators?

Response: Although there are 11 official languages in South Africa, the surrounding communities where the study was conducted were predominantly isiXhosa- and Afrikaans-speaking. The KABC-II instructions were forward and back translated in both languages and consensus of the translations were done with community members to ensure that the translations were appropriate. The test administrators were all trained psychologists with MA level degrees. We have added this information to the paper.

4) What was the rationale for testing for language at 3.5 years? This wasn't clearly stated in the manuscript. Typically, language development is divided into 3-12 months, 12-18months, 18months-2 years, 2-3 years, 3-5 years.

Response: In this prospective cohort, language was measured at multiple age points, and the timing considered the trajectory of language development and feasibility of the cohort logistics. We previously measured language at 2 years, and 3.5 years was the next time point.

5) There is not statistical testing for multiple comparisons in the risk factor analysis. Multiple risk factors compared against multiple scales on the test require this testing.

Response: We agree with the importance of considering multiple comparisons, and were careful to limit the number of tests as a way to combat this. This was an exploratory analysis and given the smaller sample size as this was in CHEU, we felt it was important to report all results. However, we have shifted the focus onto effect size estimates and report confidence intervals in order to emphasise the strength and direction of the associations. We are also more cautious in our interpretation in terms of significance, and have added to the limitations that this needs to be replicated in other settings.

Reviewer #2: This is an incredibly important area of research and I commend the authors for their work. This manuscript was clear, concise, and very enjoyable to read. I believe this work makes a meaningful, and very much needed, contribution to the academic research in this field.

Response: Thank you very much for your comments.

Introduction

Might be worth highlighting in the introduction that the majority (somewhere around 80%) of women living with HIV are of reproductive age. Intro skips from people living with HIV to children being born to HIV-infected mothers, and it might add to the argument to focus in on the large number of people living with HIV who are women of reproductive age (to further emphasize the importance of the issue the authors are examining here). 

Response: Thank you, we agree and have expanded the introduction and added in data: “The number of people living with HIV remains high, estimated at 39 million globally [5]. Adolescent girls and young women remain disproportionately affected by HIV, accounting for 60% of new HIV infections in that age group [9]. In 2022 sub-Saharan Africa, adolescent girls and young women account for over 77% of new infections among 15-24 years olds [10].”

Paragraph 2: suggested editing to “previous reviews of work in LMICs”, to clarify that the comparison between CHEU and unexposed children is all with LMICs (if this is true), or remove the LMICs framing if these meta-analyses included HICs

Response: Thank you, the meta-analyses do include HICs and therefore we have corrected this phrasing.

Specify definition of “young” for “young CHEU” (paragraph 2), or refer to all as children <24 months

Response: We agree this is non-specific and have removed the word young given we put ‘by 24 months’ later in the sentence.

Specify “older ages” in paragraph 3, e.g. children over 24 months? (since young or older could be relative), even “pre-school and school-aged children” to be a bit more specific about age

Response: We agree and have rephrased to ‘children over 24 months’ as specified.

Paragraph 3, suggested adding parameters to “understand CHEU outcomes” e.g. “to understand neurocognitive outcomes in pre-school and school-aged CHEU children”

Response: Many thanks, we have included this suggestion.

Paragraph 4, suggest adding parameters to “Several risk factors for development” e.g. “risk factors for poor neurocognitive development”

Response: We agree and have made this edit.

Paragraph 4, suggested editing “such as maternal characteristics” to “parental characteristics” (to emphasize the importance of both parents participating in caregiving for children)

Response: We agree and have made this edit.

Suggest that paragraph 4 of the introduction could be included in the discussion, around parameters outside of HIV status that intersect with the neurocognitive outcomes here, to shorten the intro a bit

Response: Thank you for this idea, we have made edits as suggested. 

Suggested editing “Drakenstein Child Health Study” to include “the study cohort is part of the Drakenstein Child Health Study”, to clarify for the reader that’s why you’re discussing this cohort, or clarify is this cohort was established specifically for this study?

Response: Thank you, we have added ‘DCHS’ to the aim to clarify this for readers: “Therefore, this analysis aimed to build upon these findings by assessing neurocognitive outcomes at 3.5 years comparing CHEU with CHUU from the DCHS, …”

Suggest reframing “harnessing the richness of the data collected to assess potential confounding variables” to “providing access to comprehensive datasets” (as “richness” could be seen as subjective and statistical methodology should determine confounding variables, vs. assessing potential)

Response: We have made these edits.

Methods

This appears to be a sub-analysis of the broader DCHS study (e.g. references 20,21), would clarify in the methods how participants were enrolled in the broader study, even overall aims of the broader study, to put this analysis in context

Response: Thank you this is correct. We have expanded the text to include the parent study recruitment and we hope that it is clearer now:

In the final paragraph of the introduction, page 4, we state: 

“Therefore, this analysis aimed to build upon these findings by assessing neurocognitive outcomes at 3.5 years comparing CHEU with CHUU from the DCHS, and investigating the contextual predictors of neurodevelopment in CHEU.”

And in the methods, page 4:

“The study cohort is part of the DCHS. Pregnant women were recruited to the DCHS between 5 March 2012 and 31 March 2015 at 20-28 weeks’ gestation at routine antenatal visits in two public healthcare clinics; mother-child dyads are being followed up in the DCHS with regular visits in addition to routine care….

At 3.5 years neurocognitive assessments were conducted of all available children.”

Clarify statement “mother-child dyads are being followed up.” e.g. prenatal visits across pregnancy followed regional recommendations etc.

Response: We apologise this was not clear. The mothers received routine prenatal and postnatal visits as well as follow up as part of the study, additional to the routine visits received as part of the national recommendations. We have clarified this as above. 

Suggested removing “so the population is relatively stable”

Response: We have done this, and edited to ‘with strong retention in study follow up.’

Keep past tense consistent (e.g. HIV prevalence in this cohort was 21%)

Response: We have edited this.

Does “only two children have HIV infection” mean from vertical transmission?

Response: Yes, we have clarified this.

Suggested editing Exposure Definition to “HIV Exposure Definition”

Response: Thank you, we have made this edit.

Could provide a bit more detail around ARV treatment (e.g. were all women receiving the same medications? Is there data on what women received which medications if there are multiple ARV regimens prescribed, what prophylactic therapy was given to children at birth? Since there is some evidence that drug exposures have impact it may be important information for the reader to have). In addition, if mothers are receiving various treatments this should likely be included as a covariate (likely too small a cohort to examine this much further).

Response: We have added in more information on maternal ART given into Table 1 as well as maternal CD4 and HIV viral load. The majority (>80% of mothers) received first-line ART. Additionally, we have examined the association of HIV-related variables with neurocognitive outcomes.

Since the children here don’t have HIV infection it’s likely more important to clarify that the neurocognitive assessment tools used here have been validated in these populations (e.g. culturally appropriate, similar geography, validation with younger cohorts (since age range is 3-18 and this cohort is all within the younger range etc.)

Response: Thank you, we have added in references for validation as per Reviewer #1’s comment above. 

Under “Domains”, would clarify if these are validated approaches to data analysis (could also be stated in the statistical methods section), or if all methods described (beyond establishing age-dependent scores) are based on what’s outlined in the KABC handbook

Response: We confirm that the NVI score and other domain scores were computed according to the KABC-II manual. We have clarified this in the methods. 

Would specify how gestational age at delivery was assessed (e.g. by ultrasound?)

Response: Thank you, we have added this to page 6: “Gestational age at delivery was assessed based on the second trimester ultrasound scan report; where these were not available, maternal report of last menstrual period dates and fundal height measurements were used.”

Under statistical analysis, could expand on what a “complete case analysis is” (e.g. is this a separate analysis of each invidual dyad?)

Response: This is a group analysis of all children who completed all three outcome assessments and had non-missing values for a priori covariates. Thus, children who did not complete all assessments were not included in the main analysis.

Statistical analysis states “children who completed all three outcome assessments”, does that mean any child who didn’t complete all assessments were removed from the analysis? Would include that if that’s the case, and/or clarify how the analysis accounted for missing data

Response: Thank you for this clarification, yes this is correct. However, we examined the distribution of covariates comparing between those children in the complete-case analysis and the full cohort. Additionally, we conducted sensitivity analyses (Supplement Table 3) including all children who completed each test and these show that the CHEU/CHUU relationships hold.

Results

Suggest removing “split between” as it might imply assignment to groups

Response: Agree, we have removed this.

Suggest removing “complete-case cohort” as could imply this is subset of the complete cohort (e.g. a case-cohort), would just refer to this as the study cohort

Response: While we understand the reviewer’s point, we still think that the term "complete-case" is useful for the reader. We have edited to mention it in the context of complete-case analysis and then just refer to the "analytic cohort" throughout.

Would it be possible to include a Figure to present the data? While the tables are very comprehensive a figure with the cognitive data might be a nice visual presentation for the reader

Response: Thank you for this suggestion. We have now added two forest plots to replace Table 2 and Table 3, which we have moved to the Supplement. 

Discussion

Suggest including some discussion around why language is impacted and no other parameters? Is there a pathobiology hypothesis there (e.g. developmental processes in utero critical to language that may be differentially impacted?), or is this an age-specific observation? Or perhaps these methodology we have available for these types of assessments is most sensitive in terms of picking up differences in language?

Response: Thank you for this comment. We have added to the discussion on page 9. Language develops rapidly in this early period. Neurodevelopment affected by HIV exposure may therefore show differences in language as this is the domain most affected, or else differences are evident earlier than other domains. The complexity of language development may also mean that this domain is most affected in younger years. We also note that language has been shown to be particularly affected in this early age group in children who are HEU [11]. Finally, language is important as it is the domain which is most highly predictive of later cognitive and academic outcomes. Given higher order cognitive function/executive functions are still developing at this early age, this further highlights the importance of follow up. 

Maybe expand on the finding that LBW was associated with poorer cognitive function in CHEU, perhaps compounding risk factors, highlighting the need to protect these children. E.g. expand on the sentence “Our findings suggest low birthweight CHEU infants may be particularly vulnerable.”

Response: We agree and have expanded as suggested on pages 10-11:

“Our findings suggest low birthweight CHEU infants may be particularly vulnerable; it is likely that the combination of risk factors (low birthweight and HIV exposure) may compound the risk of poor cognitive outcomes. Together, this research highlights the need to identify and protect those children with multiple risk factors.”

Suggest removing “large sample size”, as this could be perceived as subjective (likely could also change “small sample size” to insufficient sample size, same with “large enough sample size”, would suggest “sufficient power, or sample size”)

Response: This is a fair point and we have made this edit.

Suggest removing “unlike in many of the studies performed to date” and leave as “allowing this analysis to control for factors with an established impact on neurocognitive outcomes in children”

Response: Thank you, we agree this wording sounds better and we have made the change.

Suggested editing “this analysis has limitations” to “this study”, as limitations are not restricted to the analysis

Response: Many thanks, we have edited this.

“There were some missing outcome data across the cohort which may have resulted in selection bias.” Is stated in the discussion, should be expanded upon in the methods section. Suggest the section on missing data in the limitations paragraph of the discussion could be moved to the methods section (since it’s great to highlight this as a limitation but that’s true of essentially every study and just requires transparency in the methods on how it’s accounted for).

Response: We have added to the methods and expanded on this and have left one sentence in the limitations. 

Conclusion

Suggested rephrasing “continued subtle impairment” as subtle could be somewhat objective, perhaps consider removing it as the analysis did how an impairment

Response: We have removed this. 

Would caution the authors around this statement “The identified vulnerability of male CHEU, and low birthweight as a risk factor, may help target intervention strategies.” As it could imply sex-stratification of resource allocation, which, given female children often suffer from lower resource allocation, could send the wrong message … arguably the results in entirety show that all children who are CHEU require resource allocation (which would already be a targeted population for intervention)

Response: Many thanks for your careful consideration of our paper. We agree with this key point and have edited the statement accordingly.

Reviewer #3: It was a pleasure reviewing this manuscript that attempted to determine HIV associated neurocognitive impairment in CHEU at 3.5 years given that previous studies had covered a younger age group. There are a few comments below that need to be addressed to improve the manuscript.

Response: Thank you very much for your feedback.

Major

1. KABC raw scores were converted to scaled scores using norms from the manual and then converted to standard scores. The manual norms are from US samples. This implies that the South African sample were compared to US norms which may give biased results given cultural differences (where test items maybe more familiar to 3 to 4 year old US children than SA children). This inflates the actual rate of impaired scores and may explain why there was high rate of impairment in the CHUU of 63.5%. This high rate of impairment in a ‘control group’ should raise a red flag. What the authors can do is to use the CHUU to generate ages adjusted Z scores which can be used as the outcomes and not the scores derived from US norms. In this way, CHEU children will be compared to their our ‘norms’.

Response: Thank you for this question. We agree that there are concerns with using US norms, even in a test that is validated for this setting. We also acknowledge that this may be inflating the levels of impairment. As a result we have edited the manuscript to reduce the focus on impairment and increase focus on the between group comparison which is the main aim of the analysis. Given any limitations of using the norms will apply to both (CHEU and CHU) groups, we feel the group comparison remain valid and were most interested in the relative comparison, rather than the absolute numbers. 

We have investigated the use of z-scores and discussed various statistical approaches. However, the NVI is a composite score, defined by adding four individual components (conceptual thinking, face recognition, hand movements and triangles) which have already been scaled using US norms, and assigning a score based on the total. We were therefore concerned about validity if we created our own norms, as the age adjustment would have to be done for each of the individual components and then combining into the NVI may no longer be a valid approach. We felt that there are not equivalent validated scales for the setting, but we have conducted additional supplemental analyses where possible and highlight that new scales are needed with contextually appropriate norms in the limitations on pages 11-12. 

We hope that the focus on between-group comparisons, reduced emphasis on impairment, and addition of effect sizes have helped to alleviate this concern. Following this comment, we have also reviewed our entire analysis strategy to optimise this. As a result, we have (i) included sensitivity analyses comparing results of all children across individual tests which show the findings hold, and (2) we have modified our risk factor analyses to investigate associations stratified by sex given the findings suggesting a vulnerability in male CHEU, and reports from the literature around sex-dependent effects [12-14].

2. The high rate of impairment needs to be addressed in the discussion. What can explain these high figures in both groups?

Response: Given the comment above in relation to norms we have reduced the focus on impairment in this manuscript. As the categorisation was based on 1 standard deviation we have ensured this is termed suboptimal cognitive development. However, we note that our children come from a community experiencing multiple factors conferring development risk. The proportion of children with impairment at 3.5 years were similar to the cohort findings at 24 months, estimated at 50.5% [13] and fit with estimates of children at risk of failing to reach their developmental potential reported across sub-Saharan Africa (66%) [15]. Previously we investigated the aetiology for developmental delay including biological influences at 2 years, and work is ongoing at older ages. 

3. Memory was assessed using Atlantis which as per the KABC manual is a learning measure. There are other memory tests like Hands Movement and Number Recall which would be more appropriate.

Response: The Hand Movement and Number Recall subtests are measures of attention and, to a lesser extent, working memory. The Atlantis subtest is the only KABC-II learning subtest (learning and recognition memory) that was age-appropriate for the cohort (the other is the Rebus subtest, which is for age 5 and up).

4. The above comments if not addressed need to be included under study limitations.

Response: We have responded as above and also added to the study limitations.

Minor

1. Why do the authors emphasise ‘language’ in the title when they also state ‘neurocognition’ This appears like an over emphasis given both outcomes were assessed using the same battery. Since language was the primary focus, they could remove neurocognition (which was not associated with HIV in the study anyway) and be direct with using language only.

Response: Thank you, we debated this and do not want to overemphasise these. We have changed to ‘Language’ only. 

2. It would be helpful for the reader if the age range of the children was given in the methods of the abstract as well as the period when the study was done (word limit permitting).

Response: We have added the child age range to the abstract and main body on page 4. 

We have added to the methods: “The Drakenstein Child Health Study is a South African population-based birth cohort which enrolled in women in pregnancy with ongoing follow up”

3. In the abstract, the KABC battery used is the ‘second edition’. This should be stated to distinguish it from the original version.

Response: We have added this. 

4. The comparison group should be stated in the methods of the abstract not in the results.

Response: We corrected this. 

5. The Bayley Scales have been used more widely in Sub-Saharan Africa than the KABC-II in children 3 to 4 years old. It is not clear why the KABC was chosen instead. It does assess from 3 to 18 years, but some authors have found challenges using it in children less than 5 years due to item difficulty in younger African children. This is alluded to in the major comments above concerning the large impairment rate.

Response: Thank you, we agree about the wide use of Bayleys, however, this is validated up to the age of 42 months. We were therefore concerned of the ceiling effect at this age and chose to conduct the KABC. Further, we planned to conduct longitudinal testing at older ages and therefore it made sense to use a tool that we could repeat in the same population. We planned this as documented previously: Donald KA, Hoogenhout M, du Plooy CP, et al. Drakenstein Child Health Study (DCHS): investigating determinants of early child development and cognition BMJ Paediatrics Open 2018;2:e000282. doi: 10.1136/bmjpo-2018-000282. We do recognise the limitations of the tool which we include in the discussion, and we have expanded on these.

6. The KABC-II gives two summary scores, the NVI which the present study used and the Mental Processing Composite which is a measure of overall cognition. Why did the authors choose

the NVI over MPI, want advantages does the former possess?

Response: The NVI is recommended for non-English-speaking populations (KABC-II manual) and allows for comparisons across populations by minimizing linguistic differences. It is a useful tool in young children and we complemented with the expressive language scale.

7. This sentence in the statistics section needs to be corrected; ‘The summary statistics were shown for the full cohort and across the two levels of exposure with comparisons using two sample t-tests or Fisher’s exact tests and chi square tests for categorical and continuous data respectively.’

Response: Thank you for picking this up, we have corrected this on page 6:

“The summary statistics were shown for the full cohort and across the two levels of exposure. Comparisons between CHEU and CHUU were made using two sample t-tests for continuous data and Fisher’s exact tests and chi for categorical data respectively.”

8. The numbers in figure 1 do not add up, this needs to be corrected.

Response: Thank you, Figure 1 has been updated.

6. PLOS authors have the option to publish the peer review history of their article (what does this mean?). If published, this will include your full peer review and any attached files.

Do you want your identity to be public for this peer review? For information about this choice, including consent withdrawal, please see our Privacy Policy.

Reviewer #1: No

Reviewer #2: No

Reviewer #3: No

Response: Thank you, we have done this. 

 

References:

1. SA General Household Survey. 2019.

2. Slogrove AL, Powis KM, Johnson LF, Stover J, Mahy M. Estimates of the global population of children who are HIV-exposed and uninfected, 2000-18: a modelling study. The Lancet Global health. 2020;8(1):e67-e75.

3. Pellowski JA, Wedderburn CJ, Stadler JAM, Barnett W, Stein DJ, Myer L, et al. Implementation of prevention of mother-to-child transmission of HIV (PMTCT) guidelines in South Africa: Outcomes from a population-based birth cohort study in Paarl, Western Cape BMJ open. 2019.

4. Moyo F, Mazanderani AH, Murray T, Sherman GG, Kufa T. Achieving maternal viral load suppression for elimination of mother-to-child transmission of HIV in South Africa. Aids. 2021;35(2):307-16.

5. UNAIDS. AIDSinfo [Available from: http://aidsinfo.unaids.org.]

6. Schnoll JG, Temsamrit B, Zhang D, Song H, Ming G-l, Christian KM. Evaluating Neurodevelopmental Consequences of Perinatal Exposure to Antiretroviral Drugs: Current Challenges and New Approaches. Journal of Neuroimmune Pharmacology. 2019.

7. Cassidy AR, Williams PL, Leidner J, Mayondi G, Ajibola G, Makhema J, et al. In Utero Efavirenz Exposure and Neurodevelopmental Outcomes in HIV-exposed Uninfected Children in Botswana. The Pediatric Infectious Disease Journal. 2019;38(8):828-34.

8. World Health Organization. Consolidated guidelines on HIV prevention, testing, treatment, service delivery and monitoring: recommendations for a public health approach. 2021.

9. UNAIDS. Women, adolescent girls and the HIV response. 2020.

10. UNAIDS. 2022 [Available from: https://www.unaids.org/en/resources/fact-sheet.]

11. Wedderburn CJ, Weldon E, Bertran-Cobo C, Rehman AM, Stein DJ, Gibb DM, et al. Early neurodevelopment of HIV-exposed uninfected children in the era of antiretroviral therapy: a systematic review and meta-analysis. The Lancet Child & adolescent health. 2022;6(6):393-408.

12. Powis KM, Lebanna L, Schenkel S, Masasa G, Kgole SW, Ngwaca M, et al. Lower academic performance among children with perinatal HIV exposure in Botswana. Journal of the International AIDS Society. 2023;26(S4):e26165.

13. Donald KA, Wedderburn CJ, Barnett W, Nhapi RT, Rehman AM, Stadler JAM, et al. Risk and protective factors for child development: An observational South African birth cohort. PLOS Medicine. 2019;16(9):e1002920.

14. Bulterys MA, Njuguna I, King'e M, Chebet D, Moraa H, Gomez L, et al. Neurodevelopment of children who are HIV-exposed and uninfected in Kenya. Journal of the International AIDS Society. 2023;26(S4):e26149.

15. Black MM, Walker SP, Fernald LCH, Andersen CT, DiGirolamo AM, Lu C, et al. Early childhood development coming of age: science through the life course. Lancet. 2017;389(10064):77-90.

---

## [Editor Report · Decision Letter 1]

8 Jan 2024

Language outcomes of preschool children who are HIV-exposed uninfected: an analysis of a South African cohort

PONE-D-23-27426R1

Dear Dr. Wedderburn,

We’re pleased to inform you that your manuscript has been judged scientifically suitable for publication and will be formally accepted for publication once it meets all outstanding technical requirements.

Kind regards,

Andrea L. Conroy, PhD

Academic Editor

PLOS ONE
---

## [Editor Report · Acceptance letter]

30 Mar 2024

PONE-D-23-27426R1 

PLOS ONE

Dear Dr. Wedderburn, 

I'm pleased to inform you that your manuscript has been deemed suitable for publication in PLOS ONE. Congratulations! Your manuscript is now being handed over to our production team.

Kind regards, 

on behalf of

Dr. Andrea L. Conroy 

Academic Editor

PLOS ONE